# A C2-Domain Abscisic Acid-Related Gene, *IbCAR1*, Positively Enhances Salt Tolerance in Sweet Potato (*Ipomoea batatas* (L.) Lam.)

**DOI:** 10.3390/ijms23179680

**Published:** 2022-08-26

**Authors:** Chang You, Chen Li, Meng Ma, Wei Tang, Meng Kou, Hui Yan, Weihan Song, Runfei Gao, Xin Wang, Yungang Zhang, Qiang Li

**Affiliations:** Xuzhou Institute of Agricultural Sciences in Jiangsu Xuhuai District/Sweetpotato Research Institute, Chinese Academy of Agricultural Sciences/Key Laboratory of Biology and Genetic Breeding of Sweetpotato, Ministry of Agriculture and Rural Affairs, Xuzhou 221131, China

**Keywords:** sweet potato, *IbCAR1* gene, salt stress resistance, abscisic acid

## Abstract

Plant C2-domain abscisic acid-related (CAR) protein family plays an important role in plant growth, abiotic stress responses, and defense regulation. In this study, we cloned the *IbCAR1* by homologous cloning method from the transcriptomic data of Xuzishu8, which is a sweet potato cultivar with dark-purple flesh. This gene was expressed in all tissues of sweet potato, with the highest expression level in leaf tissue, and it could be induced by NaCl and ABA. Subcellular localization analyses indicated that IbCAR1 was localized in the nucleus and plasma membrane. The PI staining experiment revealed the distinctive root cell membrane integrity of overexpressed transgenic lines upon salt stress. Salt stress significantly increased the contents of proline, ABA, and the activity of superoxide dismutase (SOD), whereas the content of malondialdehyde (MDA) was decreased in overexpressed lines. On the contrary, RNA interference plants showed sensitivity to salt stress. Overexpression of *IbCAR1* in sweet potatoes could improve the salt tolerance of plants, while the RNAi of *IbCAR1* significantly increased sensitivity to salt stress in sweet potatoes. Meanwhile, the genes involved in ABA biosynthesis, stress response, and reactive oxygen species (ROS)-scavenging system were upregulated in overexpressed lines under salt stress. Taken together, these results demonstrated that *IbCAR1* plays a positive role in salt tolerance by relying on the ABA signal transduction pathway, activating the ROS-scavenging system in sweet potatoes.

## 1. Introduction

The growth of plants is constantly affected by various environmental factors, such as low temperature, drought, salinity, heavy metal, and other abiotic stresses [1,2]. Salt stress will lead to a decrease in the yield and quality of sweet potatoes [3,4,5]. Therefore, one of the important objectives of sweet potato breeding in China is to find salt tolerance genes and select sweet potato varieties with strong resistance to stresses [6,7,8,9,10]. The development of genetic engineering technology provides a new idea and way for breeding sweet potato varieties with high yield, high quality, and strong resistance to stresses [11,12].

The C2 domain is a relatively conserved domain with 130 residues located between the C1 domain and the catalytic domain of protein kinase C (PKC) [13,14]. Compared with C2 domain proteins in animals, plant C2 domain proteins contain only one C2 domain and no other conserved domains. Therefore, plant C2 domain proteins are also known as small C2 domain proteins. Proteins containing C2 domains play important roles in many biological processes [15]. To date, extensive evidence has shown that genes encoding plant C2 domain proteins are involved in the regulation of plant abiotic stress resistance, such as high salt and drought stress [16]. In *Arabidopsis*, CAR (C2-domain abscisic acid-related) proteins respond to salt stress by mediating ABA signaling [17]. Pepper soybean genes regulated by cold 2 (SRC2) protein are involved in the resistance of bacteria, pathogens, and abiotic stress [18]. *Triticum aestivum* elicitor responsive gene (*TaERG3*) encoding wheat C2 domain protein can be induced by low temperature, high salt, and stripe rust [19]. Glycine max C2 domain gene (*GMC2-148*) could increase proline content, decrease malondialdehyde (MDA) content, and upregulate the expression of *NAM*-*ATAF1/2*-*CUC2* (*NAC11*) gene, cold-responsive (*COR47*) gene, cold-inducible (*KIN1*) gene and other abiotic transcription factors in soybean leaves, thus improving the tolerance of salt and drought stress [20]. Therefore, an in-depth analysis of C2 domain proteins in plants and their regulatory mechanisms in flower morphogenesis, biological stress, and abiotic stress is of great significance for genetic improvement in crop yield, quality, and resistance [21,22].

Sweet potato (*Ipomoea batatas* (L.) Lam.) is an important crop, feed, and industrial raw material crop [23,24,25]. Its productivity is often restricted by salt stress. Cultivated sweet potato is an autohexaploid crop with a large number of chromosomes (2n = 6x = 90), complex genetic background, and incompatibility between intraspecific and interspecific crosses, resulting in a lack of available germplasm resources. It is difficult to cultivate sweet potato varieties with excellent comprehensive traits by using traditional crossbreeding methods [1,26]. Genetic engineering is an effective way to improve the salt tolerance of sweet potatoes [27,28,29,30,31,32]. However, there are few reports about CAR conferring tolerance to salt in sweet potatoes [33,34,35]. In this study, we cloned the stress-resistant gene *IbCAR1* by homologous cloning method from the transcriptomic data of Xuzishu8, which is a sweet potato cultivar with dark-purple flesh. Its overexpression enhanced salt tolerance in transgenic sweet potatoes.

## 2. Results

### 2.1. Cloning and Sequence Analysis of IbCAR1 and Its Promoter

The open reading frame (ORF) of the *IbCAR1* gene is 531bp, and 176 amino acids are encoded (Appendix A). The molecular weight (MW) of IbCAR1 protein is 19.95 kDa, and the isoelectric point (pI) is 7.76. This protein has a highly conserved C2 domain (Figure 1A). It has a high sequence identity with CAR in *Ipomoea trifida* (100%), *Ipomoea triloba* (XP_031110439.1, 100%), *Ipomoea nil* (XP_019173082.1, 94.32%), *Capsicum annuum* (XP_016560847.1, 74.40%), *Hevea brasiliensis* (XP_021664250.1, 74.40%), *Sesamum indicum* (XP_011073630.1, 73.81%), *Ipomoea nil* (XP_019182458.1, 73.30%), *Coffea eugenioide* (XP_027161458.1, 73.21%), *Datura stramonium* (MCD9559783.1, 71.43%), *Solanum pennellii* (XP_015065130.1, 70.83%), *Camellia sinensis* (XP_028068988.1, 70.83%), *Solanum lycopersicum* (XP_004231984.1, 70.24%), *Impatiens glandulifera* (XP_047317802.1, 69.64%), *Tripterygium wilfordii* (XP_038689876.1, 69.64%), *Mangifera indica* (XP_044467456.1, 69.05%), and *Eucalyptus grandis* (XP_010032690.1, 67.86%) (Figure 1B). Phylogenetic analysis showed that IbCAR1 has a close relationship with that of *Ipomoea trifida* (Figure 1C). The 2018-bp genomic DNA of *IbCAR1* contained three exons and two introns (Figure 1D). A 1440bp fragment corresponding to the promoter of *IbCAR1* was isolated from Xuzishu8 genomic DNA and analyzed by using the online analysis software PlantCARE. The results showed that the promoter possessed not only basic *cis*-acting elements but also photoresponsive elements and *cis*-acting elements related to ABA, salicylic acid, and protein binding (Appendix A).

### 2.2. Expression Patterns of IbCAR1 Genes

To investigate the expression level of *IbCAR1* in different tissues of Xuzishu8, including the storage root, pencil root, fibrous root, leaf, stem, and petiole. The IbCAR1 gene was expressed in different tissues of sweet potato, but the highest expression was in leaves and the lowest in petioles (Figure 2A). To further analyze its potential function, the expression of *IbCAR1* was checked using the whole plants of Xuzishu8, which were treated with 200 mM NaCl and 100 mM ABA for 0, 3, 6, 12, 24, and 48 h. These results showed that the expression of *IbCAR1* was significantly induced by NaCl and ABA. Under 200 mM NaCl stress, the expression of *IbCAR1* peaked at 12 h with 2.93-fold, respectively. Under ABA stress, the expression of the *IbCAR1* gene increased sharply after 24 h (Figure 2B).

### 2.3. Subcellular Localization of IbCAR1

The subcellular localization of *IbCAR1* was studied by detecting IbCAR1-GFP fusion protein. The ORF of *IbCAR1* controlled by the 35S promoter was fused with GFP to construct the subcellular localization vector pCAMBIA1300-IbCAR1-GFP. The pCAMBIA1300-IbCAR1-GFP was expressed in *Nicotiana benthamiana* leaf epidermal cells using *Agrobacterium tumefaciens* (*A. tumefaciens*)-meditated transformation. Confocal scanning microscopic images from *Nicotiana benthamiana* leaf epidermal cells showed that the IbCAR1-GFP fusion expression protein was located in the nucleus and plasma membrane (Figure 3).

### 2.4. Determination of Plasma Membrane Integrity of Transgenic Sweet Potato under Salt Stress

To further investigate whether *IbCAR1* contributes to salt resistance, we generated three overexpressed transgenic lines (OE-1, OE-5, and OE-16) (Appendix A) and two *IbCAR1* RNAi transgenic lines (RNAi-10 and RNAi-11) (Appendix A) of sweet potato. The expression level of *IbCAR1* in overexpressed transgenic plants was significantly higher than that of WT, while the expression levels of RNAi plants were lower than that of WT. The transgenic and wild-type Xuzishu8 seedlings were hydroponic for 3 days and treated with 150 mmol·L^−1^ NaCl, while the control group was treated with water. After 24 h, PI staining was performed on the sweet potato root tip to observe the integrity of the plasma membrane of the sweet potato root cell. The results indicated that the cytoplasmic membranes in the meristem area of the root tip of sweet potato in the control group showed weak red fluorescence, indicating that the cytoplasmic membranes were intact. After being treated with salt, the root elongation zone (2–3 mm from the tip) of the WT showed strong red fluorescence in the nucleus of most cells, indicating that the plasma membrane integrity in this root region was damaged. However, the PI-stained nucleus in the same position of the overexpressed transgenic lines was substantially smaller than that of the WT under saline conditions. These results showed that WT is more sensitive to salinity stress than the overexpressed transgenic lines. In order to further verify the integrity of the root cell plasma membrane after salt stress treatment, PI staining results of wild-type Xuzishu8 and RNAi transgenic sweet potato under the same conditions were observed. After being treated with salt, the root elongation zone of the RNAi transgenic lines showed stronger red fluorescence in the nucleus of most cells, indicating that the plasma membrane integrity in this root region was completely destroyed (Figure 4). The PI staining experiment revealed the distinctive root cell membrane integrity of overexpressed transgenic lines upon salt stress. These results suggest that the *IbCAR1* gene can better maintain the integrity of the root cell plasma membrane under salt stress.

### 2.5. Overexpression of IbCAR1 Enhances Salt Tolerance in Sweet Potato

The overexpressed transgenic lines, RNAi transgenic lines, and wild-type Xuzishu8 were treated with 150 mmol·L^−1^ NaCl, and the control group was cultured with water for 3 days. Phenotypic identification showed that there was no significant difference among the overexpressed transgenic lines, RNAi lines, or WT under normal growth conditions. After the salt treatment, the growth of all sweet potato plants was inhibited to some extent. The leaves of the RNAi plants turned yellow and were more wilted than the WT plants. In the overexpressed transgenic lines, only a few bottom leaves showed yellowing and wilting, which indicated significantly increased salt tolerance (Figure 5A). It indicated that overexpression of *IbCAR1* improved the salt stress tolerance in sweet potatoes.

Salt stress can reduce the scavenging function of intracellular ROS. Accumulation of ROS will lead to membrane lipid peroxidation, MDA formation, and activation of the ROS-scavenging system. At the same time, plant cells will accumulate a large amount of proline under salt stress to maintain the normal swelling pressure of cells and improve the tolerance of plants to salt stress. The content of MDA, proline, ABA, and SOD in the leaves of overexpressed transgenic lines and WT were determined under normal and salt treatment conditions. The results indicated that the overexpressed lines increased the content of ABA and proline, increased activities of SOD, and decreased the content of MDA than that in WT plants under salt stress (Figure 5B–E). To further verify whether there is a necessary functional link between salt tolerance and *IbCAR1* gene expression, we also measured the content of MDA, proline, ABA, and SOD of RNAi lines and WT lines under normal and salt treatment conditions. The results indicated that the RNAi lines decreased the content of ABA and proline, decreased activities of SOD, and increased the content of MDA than that in WT plants under salt stress (Figure 5F–I). This is the opposite of overexpressed transgenic lines. These results all indicated that *IbCAR1* plays a role in improving the tolerance of salt stress.

### 2.6. Overexpression of IbCAR1 Upregulates the Expression of the Stress-Responsive Genes

To investigate the reason that *IbCAR1* affected salt tolerance in transgenic plants, we analyzed the expression of several genes involved in different pathways. Our data indicated that after 12 h of the salt treatment, the expression levels of the ROS scavenging-related genes *IbAPX*, *IbPOD*, *IbCAT* (encoding an ascorbate peroxidase, a peroxidase, a catalase, respectively), the late embryogenesis abundant gene *IbLEA* (encoding a late embryogenesis abundant protein), the ABA biosynthesis-related genes *IbAAO* and *IbABA2* (encoding an ascorbic acid oxidase, a zeaxanthin epoxidase, respectively) were significantly upregulated in the overexpressed transgenic plants compared with WT under salt stress (Figure 6A).

To further verify whether there is a necessary functional link between salt tolerance and *IbCAR1* gene expression, qRT-PCR was used to detect the resistance-related gene expression levels of *IbCAR1* RNAi transgenic lines and WT lines. Under normal growth conditions, the expression levels of *IbAPX* and *IbABA2* were higher in RNAi lines compared to WT, while *IbPOD*
*and*
*IbLEA* were lower in RNAi lines. For salt stress, the expression levels of *IbAPX*, *IbPOD*, and *Ib**CAT* were significantly downregulated in RNAi plants, which suggests that *IbCAR1* can regulate ROS-scavenging pathways. Meanwhile, the expression levels of *IbAAO* and *IbABA2* were significantly downregulated in RNAi plants (Figure 6B). This is the opposite of the overexpressed transgenic lines. These results indicated that *IbCAR1* may regulate stress tolerance by regulating the expression of salt stress-related genes.

## 3. Discussion

### 3.1. Overexpression of IbCAR1 Upregulates Related Genes in ABA Signaling Pathway

The plant CAR protein family plays an important role in regulating abiotic and biological stress [16]. However, so far, there are few reports about the plant CAR protein family involved in salt stress. Studies on the plant CAR protein family have focused on *Arabidopsis thaliana*. In *Arabidopsis thaliana*, CAR proteins respond to salt stress by mediating ABA signaling. Overexpression of rice CAR homolog *OsSMCP1* can improve salt tolerance and drought tolerance in *Arabidopsis thaliana*. In previous studies, CAR proteins were found to positively regulate ABA signaling by affecting the subcellular localization of PYR/PYL. CAR proteins have a molecular mechanism that integrates Ca^2+^ signal recognition, plasma membrane interaction, and PYR/PYL receptor recognition and interaction. *LOT1* affects ABA signaling by regulating the stability and dynamic localization of *CAR9*, thereby enhancing plant tolerance to drought stress [32].

ABA is an important plant hormone that regulates plant growth and development [36,37]. The ABA signaling pathway can regulate ABA-dependent stress response genes, and increasing ABA content can upregulate the expression of related stress response genes, thus enhancing the adaptability of plants to abiotic stress [38,39,40]. ABA biosynthesis can be induced by drought and salt stress [41,42,43]. It has been reported that *ABA2* and *AAO* are responsible for ABA accumulation [44,45]. Under abiotic stress, genes related to the ABA signaling pathway *AAO* and *ABA2* were induced to express and promote the rapid accumulation of ABA in plants [46]. ABA has been reported to regulate the expression levels of stress-tolerance-related genes, including *APX*, *POD*, *LEA*, and *CAT*, in several plant species [47,48,49].

In previous studies, CAR proteins have been found to participate in ABA-dependent signaling pathways by recruiting ABA signaling receptor PYR/PYL. In this experiment, genes related to the ABA signal transduction pathway *AAO* and *ABA2* and the content of ABA significantly increased in the overexpressed transgenic plants compared with WT under salt stress (Figure 5 and Figure 6). The expression levels of *POD*, *LEA*, and *CAT* were significantly upregulated in the overexpressed transgenic plants. At the same time, the expression of genes related to stress tolerance in RNAi transgenic lines was opposite to that in overexpressed transgenic lines (Figure 6). These results indicated that *IbCAR1* might play an important role in the regulation of the stress-responsive gene via the ABA signaling pathway (Figure 7).

### 3.2. Overexpression of IbCAR1 Improve Salt Tolerance in Sweet potato

Sweet potato is an important food, feed, and new energy crop, but due to the large number of chromosomes (2n = 6x = 90), complex genetic background, and incompatibility between intraspecific and interspecific hybridization, it is difficult to breed sweet potato varieties with excellent comprehensive traits by traditional crossbreeding methods, so genetic engineering is used to improve sweet potato [26]. At present, there is no related study on this protein in sweet potatoes. In this study, the gene *IbCAR1* related to acquiring stress resistance was cloned according to the transcriptome data of Xuzishu8 obtained earlier. *IbCAR1* was expressed in all tissues of sweet potato, with the highest expression level in leaves. Its expression was induced by NaCl and ABA (Figure 2A,B).

The cell membrane is a barrier separating cells from the external environment, an important structure to ensure normal internal physiological functions, and an important target for damage caused by environmental stress [50]. PI staining can show the integrity of the plasma membrane in the root elongation zone (2–3 mm from the tip). After salt treatment, the PI-stained nucleus in the root elongation zone of the overexpressed transgenic lines was substantially smaller than that of the WT, while the PI-stained nucleus in the same position of the RNAi transgenic lines showed more intense red fluorescence (Figure 4). These results suggest that the *IbCAR1* gene can better maintain the integrity of the root cell plasma membrane under salt stress.

### 3.3. Overexpression of IbCAR1 Enhances the ROS-Scavenging System

Under salt stress, a large number of ROS were produced in sweet potatoes, which caused oxidative damage to cell structure and function [51,52]. To avoid ROS damage, plants have evolved a sophisticated ROS clearance system to protect them from ROS [53]. SOD is closely related to the clearance of reactive oxygen species and catalyzes the conversion of superoxide to oxygen and hydrogen peroxide, which can be removed by *POD*, *CAT,* and *APX* [54,55]. The upregulated expression of related stress response genes *POD*, *CAT,* and *APX* enhanced their ability of ROS scavenging so as to resist the damage of ROS to plants.

In this study, the activity of SOD, the content of proline, and ABA were significantly increased, while the content of MDA significantly decreased in overexpressed plants compared with WT under salt stress (Figure 5B–E). In contrast, the physiological indicators in RNAi transgenic lines were opposite to that in overexpressed transgenic lines under salt stress (Figure 5F–I). The expression levels of *IbAPX*, *IbPOD*, *IbLEA*, *IbCAT*, *IbAAO*, and *IbABA2* in overexpressed plants were significantly upregulated, while the expression levels of *IbAPX*, *IbPOD*, *IbLEA*, *Ib**CAT*, *IbAAO*, and *IbABA2* in RNAi plants were significantly downregulated (Figure 6). These results indicated that the *IbCAR1* gene can improve the salt tolerance of transgenic sweet potato plants by upregulating the expression of stress-responsive genes and enhancing the activity of ROS-scavenging enzymes.

## 4. Materials and Methods

### 4.1. Plant Materials and Growth Conditions

The function of *IbCAR1* was studied by using Xuzishu8 (purple skin and purple flesh, table use type) as materials. Sweet potato seedlings were planted in Xuzhou’s modern agricultural experiment and demonstration base in Jiangsu province for conventional management.

### 4.2. Cloning and Sequence Analysis of IbCAR1

Total RNA was extracted from the leaves of Xuzishu8 using the RNAprep Pure Plant Kit (Tiangen Biotech, Beijing, China), and the first strand cDNA synthesis was synthesized using ReverTra Ace^®^ qPCR RT Master Mix with gDNA Remover Synthesis Kit (Toyobo, Osaka, Japan). According to the EST obtained in a previous study and referring to the genomic data of *Ipomoea trifida* (http://sweetpotato.uga.edu/, accessed on 10 November 2020), the cDNA sequence of the *IbCAR1* gene was obtained with a primer pair (*IbCAR1*-clon-F/R) and homology-based cloning method. All special primers are listed in Appendix A. The MW and pI of IbCAR1 protein were predicted using Expasy (https://web.expasy.org/protparam/ accessed on 25 November 2020). *IbCAR1* was analyzed with online BLAST (https://blast.ncbi.nlm.nih.gov/Blast.cgi accessed on 3 December 2020). COBALT was used for homology analysis, and MEGA 7.0 was used to construct an evolutionary tree (https://www.megasoftware.net/ accessed on 12 December 2020). The structure of the *IbCAR1* gene was constructed using GSDS 2.0 (http://gsds.gao-lab.org/ accessed on 5 January 2021). The promoter sequence of the *IbCAR1* gene was analyzed using the PlantCARE database (https://www.dna.affrc.go.jp/PLACE/?action=newplace accessed on 8 March 2021).

### 4.3. Expression Analysis of IbCAR1

The expression levels of the *IbCAR1* gene in six tissues (the storage root, pencil root, fibrous root, leaf, stem, and petiole) of the Xuzishu8 field seedling were determined. Xuzishu8 seedlings with the same growth pattern of 25–30 cm were cut and treated with 200 mM NaCl and 100 mM ABA, respectively. The whole plants were sampled at 0 h, 3 h, 6 h, 12 h, 24 h, and 48 h after treatment. qRT-PCR was used to analyze the expression of the *IbCAR1* gene at different time points under different stress treatments.

### 4.4. Subcellular Localization

The ORF of the terminating codon *IbCAR1* gene was removed, and the restriction endonuclease *StuⅠ* and *XbaⅠ* were used to cut the fragment into pCAMBIA1300-GFP to construct the subcellular localization vector pCAMBIA1300-IbCAR1-GFP. The recombinant vectors pCAMBIA1300-IbCAR1-GFP and pCAMBIA1300-GFP (as control) were transformed into *A. tumefaciens* strain EHA105 by heat shock method, respectively, and transiently expressed in *Nicotiana benthamiana* leaf epidermal cells using Agrobacterium infiltration [56]. After 2–3 days of low light culture, the fluorescence of the lower epidermis of injected tobacco was observed by laser confocal microscope (Nikon C2).

### 4.5. Production of Transgenic Sweet Potato Plants

The coding region of *IbCAR1* was inserted into the pCAMBIA1301S expression vector. A pair of forward and reverse nonconserved fragments of *IbCAR1* were inserted into the plant RNA interference (RNAi) vector pFGC5941. Vectors were individually introduced into Xuzishu8 via *A. tumefaciens*-mediated transformation as previously described. The resistant calli in suitable condition on the screening medium were transferred to an MS medium containing 1 mg/L ABA and 200 mg/L Cef, and the light intensity (2500 Lux) induced the formation of somatic embryos. Mature somatic embryos were cultured for 1–2 months to obtain transgenic plants [57]. The putative transgenic sweet potato plants were identified by PCR analysis with qRT-PCR primers (Appendix A).

### 4.6. Determination of Plasma Membrane Integrity

The transgenic and wild-type Xuzishu8 seedlings were treated with 150 mmol·L^−1^ NaCl, while the control group was treated with water. After 24 h, the plasma membrane integrity in root cells was checked by using propidium iodide (PI) staining [58]. Root tips (3 cm) were collected from non-treated or NaCl-treated WT overexpressed transgenic lines and RNAi transgenic lines and were incubated in a staining buffer containing 5 mM KCl/MES and 3 μg mL^−1^ PI (Life Technologies, Carlsbad, CA, USA) for 20 min. The samples were then washed in KCl/MES buffer for 5 min before imaging (elongation root zone) with an Olympus BX 63 epifluorescence microscope (Olympus, Tokyo, Japan).

### 4.7. Assay for Salt Tolerance

The overexpressed transgenic lines, RNAi transgenic lines, and wild-type xuzishu8 were treated with 150 mmol·L^−1^ NaCl, and the control group was cultured in water for 3 days for phenotypic identification. Meanwhile, proline and MDA contents and SOD activity in the leaves of transgenic and WT plants were analyzed using Assay Kits [59] (Solarbio Science & Technology Co., Ltd. Beijing, China). The ABA content in the leaves of transgenic and WT plants was analyzed using Assay Kits (Enzyme-linked Biotechnology Co., Ltd., Shanghai, China).

### 4.8. Expression of Salt Stress-Responsive Genes

The overexpressed transgenic lines, RNAi transgenic lines, and WT were treated with 150 mmol·L^−1^ NaCl, and the control group was cultured in water for 12 h. qRT-PCR was used to detect the expression levels of related resistance genes in the leaves of wild-type and transgenic plants under salt stress with gene-specific primers [60] (Appendix A).

### 4.9. Statistical Analysis

SPSS 22.0 software (IBM, New York, NY, USA) was used for statistical analysis of data, and *t*-test analysis of variance was used between different samples. All experiments were performed with three biological replicates. The two-tailed *t*-test was used for significant difference analysis.

## 5. Conclusions

The stress-resistant gene *IbCAR1* was obtained by homologous cloning. This is the first report that *IbCAR1* endows sweet potatoes with salt tolerance. The overexpression of the *IbCAR1* gene relies on the ABA signal transduction pathway, regulating the expression of stress-related genes and activating the ROS-scavenging system; thus, the salt tolerance of transgenic sweet potato plants was significantly improved. This study provides a new candidate gene for improving the salt tolerance of sweet potatoes.

## Figures and Tables

**Figure 1 ijms-23-09680-f001:**
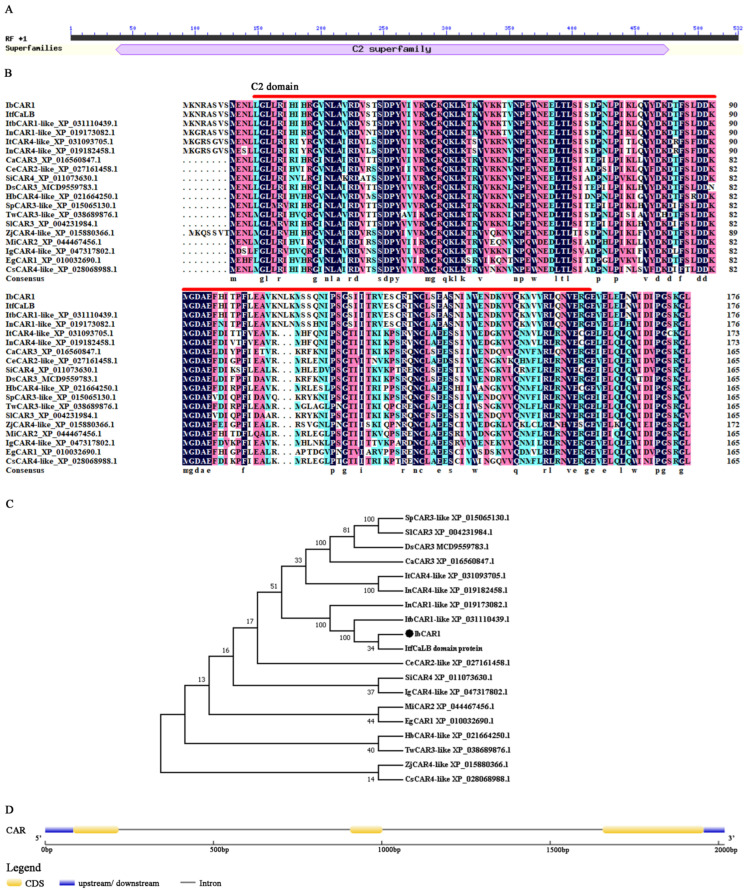
Sequence analysis of IbCAR1. (**A**) Features of IbCAR1 protein. (**B**) Sequences alignment of IbCAR1 with its closest homologs from other species. The C2 domain is represented with red lines. (**C**) Analysis of IbCAR1 protein homology tree in different species. (**D**) The structure diagrams of *IbCAR1.* Exons are represented by ellipses, and introns are represented by lines. The blue boxes represent the 5′ and 3′ untranslated regions (UTR).

**Figure 2 ijms-23-09680-f002:**
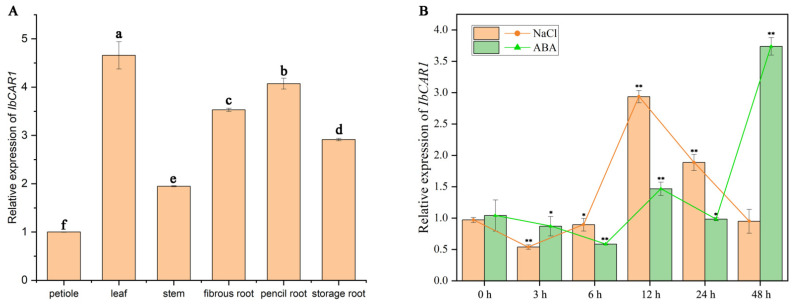
Expression analysis of *IbCAR1*. (**A**) Expression analysis of *IbCAR1* in storage root, pencil root, fibrous root, leaf, stem, and petiole tissues of Xuzishu8. Data are presented as means ± SE (*n* = 3). Different lowercase letters indicate a significant difference at *p* < 0.05 based on Student’s *t*-test. (**B**) Expression analysis of *IbCAR1* in whole plants of Xuzishu8 after different times (h) in response to 200 mM NaCl and 100 mM ABA, respectively. The sweet potato *b-actin* gene was used as an internal control. Data are presented as means ± SE (*n* = 3). ** indicates a significant difference compared with 0 h (*p* < 0.01) based on Student’s *t*-test. * indicates a significant difference compared with 0 h (*p* < 0.05) based on Student’s *t*-test.

**Figure 3 ijms-23-09680-f003:**
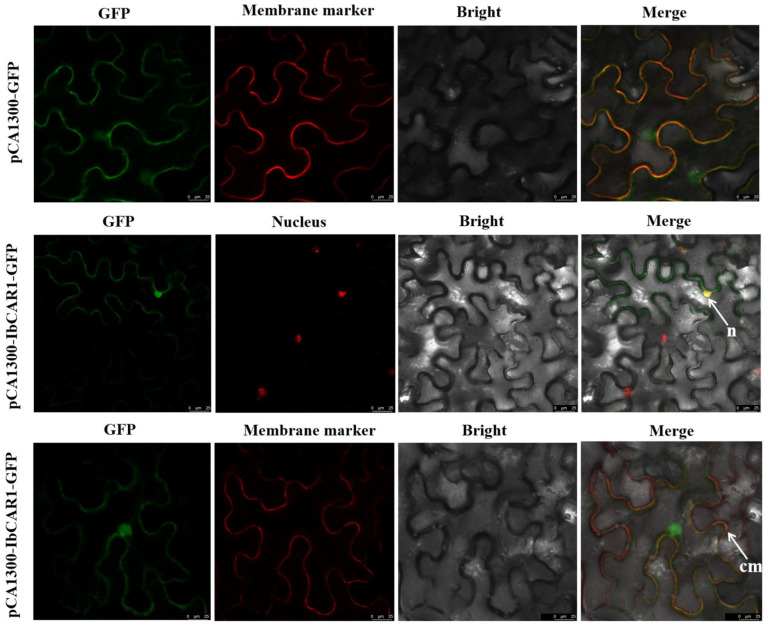
Subcellular localization of IbCAR1. *OsMADS53* and *AtSYP122* were used as nuclear and membrane markers, respectively. The n stands for nucleus. The cm stands for cytoplasmic membrane. Confocal scanning microscopic images showed that the IbCAR1-GFP fusion expression protein localized on the nucleus and membrane. GFP as the control. Bars = 25 μm.

**Figure 4 ijms-23-09680-f004:**
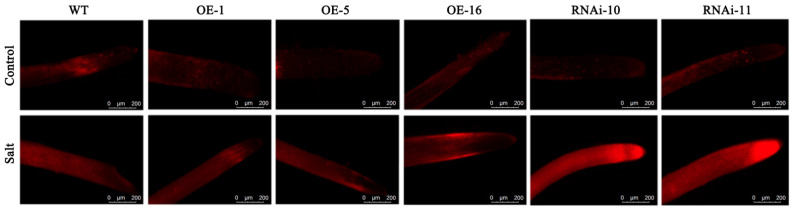
Effects of NaCl stress on the root cell membrane integrity. The plasma membrane integrity in root cells was checked by using propidium iodide (PI) staining. The PI staining experiment revealed the distinctive root cell membrane integrity of overexpressed transgenic lines upon salt stress. The PI staining image of the root elongation zone (three independent experiments) for each treatment. Bars = 200 μm.

**Figure 5 ijms-23-09680-f005:**
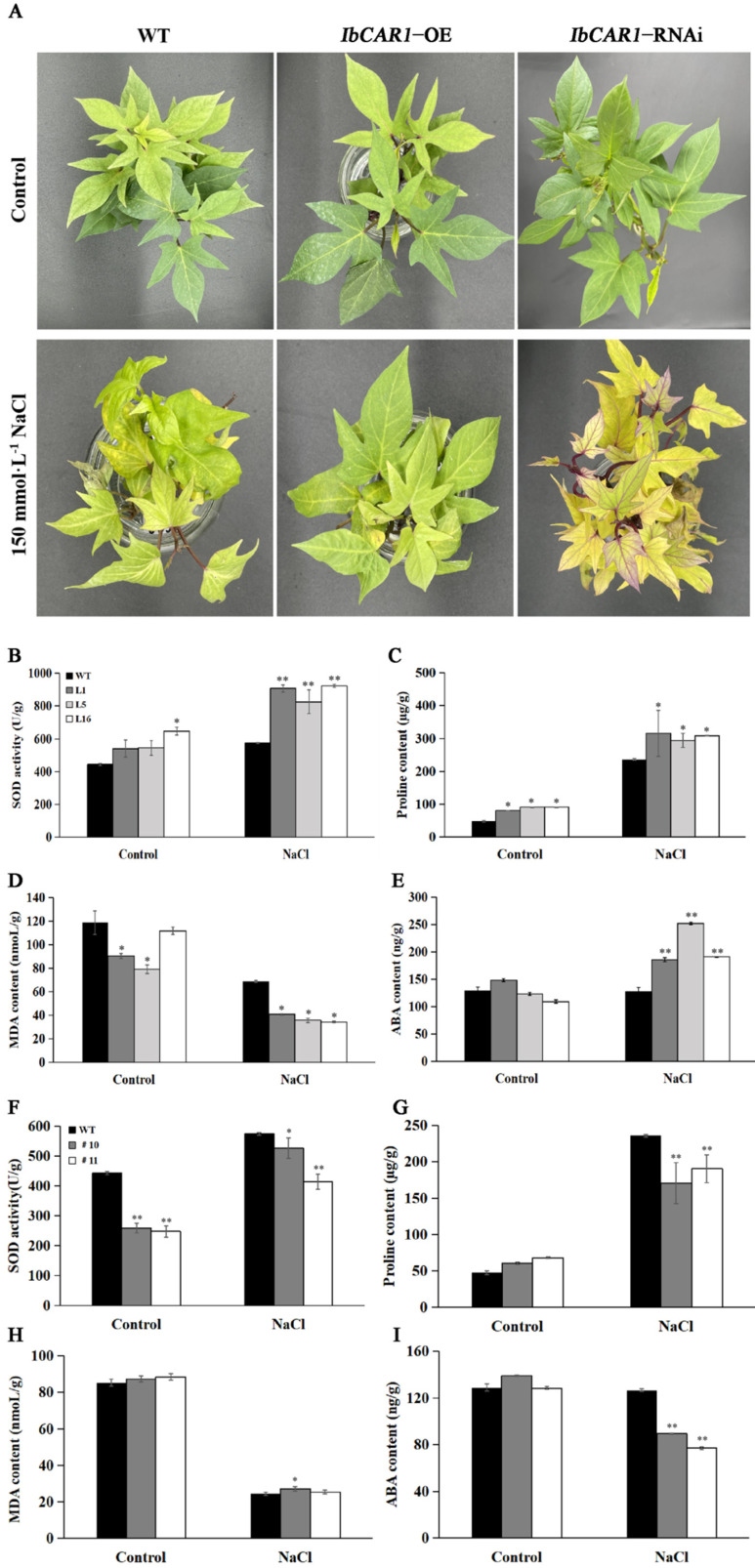
Analysis of the function of sweet potato *IbCAR1* under normal and salt stress. (**A**) Phenotypic analysis of WT and transgenic sweet potato under salt stress. (**B**–**E**) The SOD (**B**), Proline (**C**), MDA (**D**), and ABA (**E**) contents of WT and overexpressed plant leaves under normal and salt stress. (**F**–**I**) The SOD (**F**), proline (**G**), MDA (**H**), and ABA (**I**) contents of WT and RNAi plant leaves under normal and salt stress. Data are presented as means ± SE (*n* = 3). ** and * indicate significant differences between the transgenic lines and WT at *p* < 0.01 and *p* < 0.05 levels, respectively.

**Figure 6 ijms-23-09680-f006:**
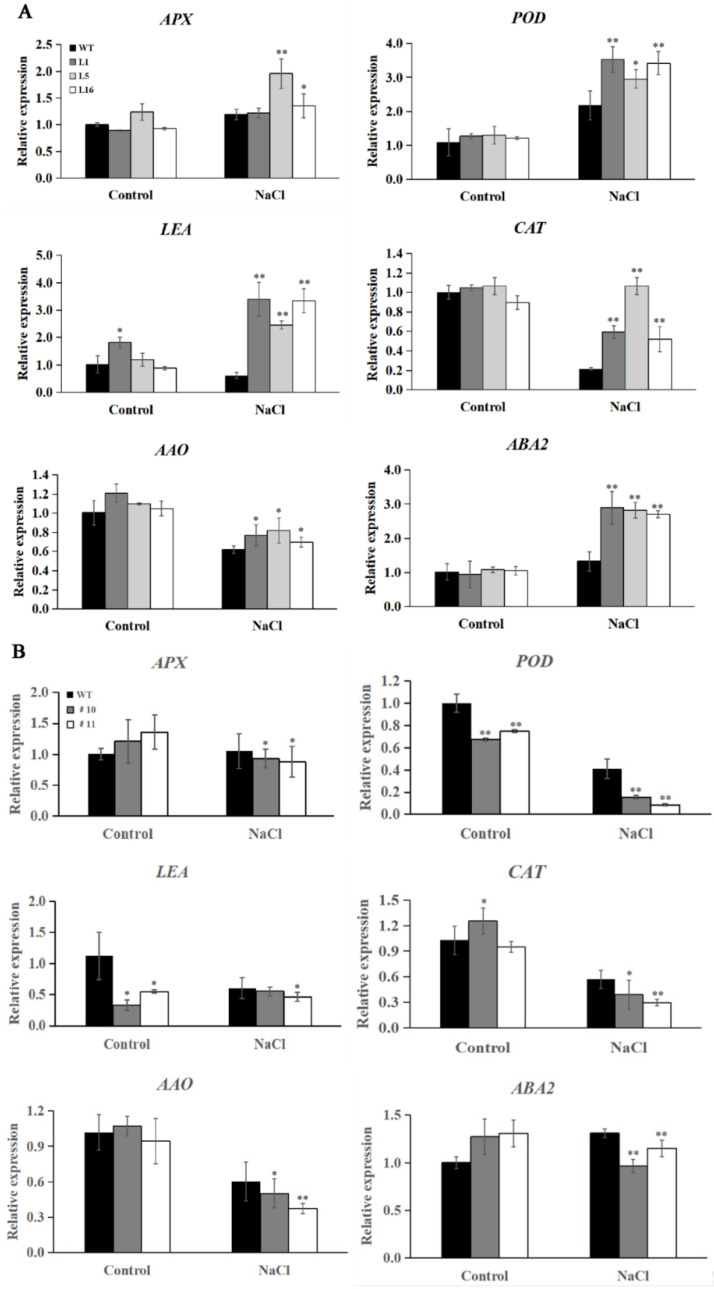
Expression levels of stress-responsive genes in transgenic and WT sweet potato plants. (**A**) Expression patterns of stress-related genes in the WT and the overexpressed transgenic lines L1, L5, and L16. (**B**) Expression patterns of stress-related genes in the WT and the RNAi transgenic lines #10, #11. Plants grown in the transplanting boxes were sampled for analysis after treating with no stress (normal) for 12 h and salt stress for 12 h. Data are presented as means ± SE (*n* = 3). ** and * indicate significant differences between the transgenic lines and WT at *p* < 0.01 and *p* < 0.05 levels, respectively.

**Figure 7 ijms-23-09680-f007:**
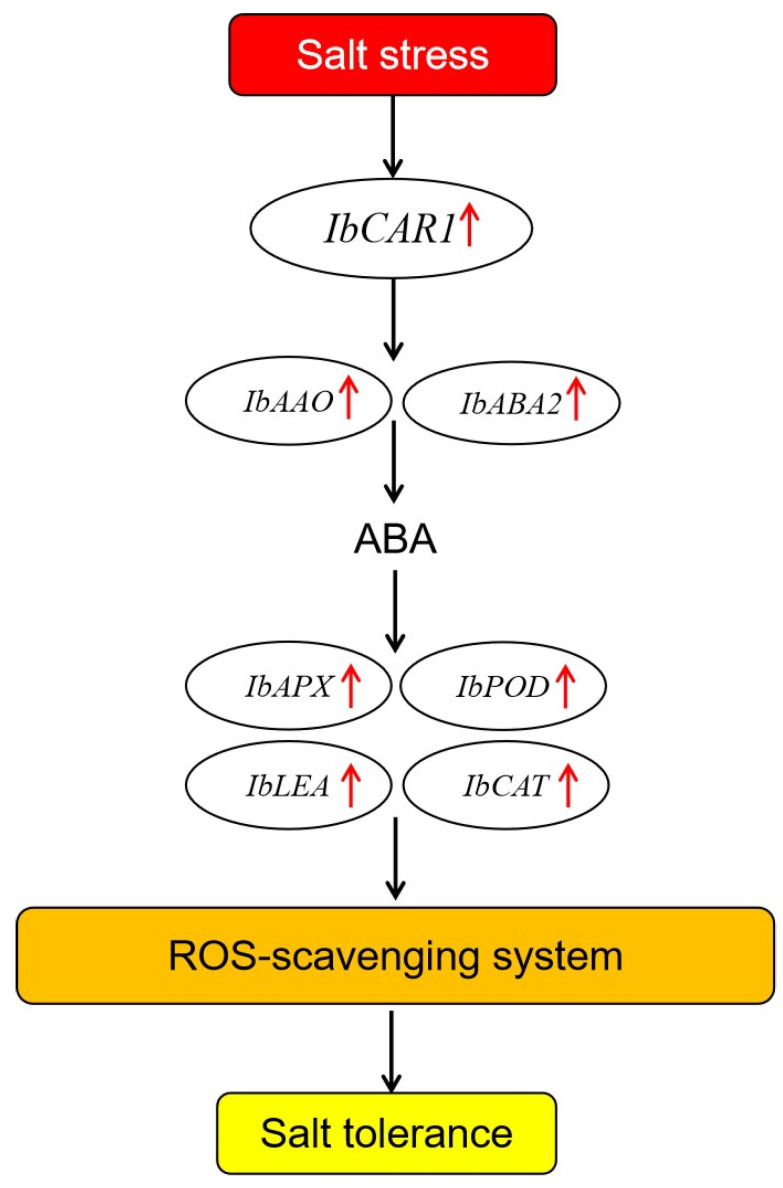
A proposed model for regulation of IbCAR1 in salt stress tolerance in transgenic sweet potato. Red arrows indicate up-regulation of gene coding these enzymes.

## Data Availability

The data presented in this study are available on request from the corresponding author.

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
