# Peer review of "A C2-Domain Abscisic Acid-Related Gene, IbCAR1, Positively Enhances Salt Tolerance in Sweet Potato (Ipomoea batatas (L.) Lam.)"

_ijms, 2022, doi:10.3390/ijms23179680_

Round 1

Reviewer 1 Report

The authors cloned IbCAR1 gene by homologous cloning method from the transcriptomic data of Xuzishu8 and investigated the expression levels and subcellular localization of this gene. Based on the experimental results using the transgenic sweetpotato plants (overexpressed and RNAi), they suggested that IbCAR1 gene plays a positive role in salt tolerance through ABA signaling pathway, activating the ROS scavenging system in sweetpotato. I think this study provides new insight into IbCAR1 gene and its molecular function in sweetpotato. However, I would like to suggest some modifications to improve this manuscript.

  1. Introduction (line: 56)

The authors used the word ‘grain’ for sweetpotato. I think ‘crop’ is better for sweetpotato.

  1. Results (lines: 134-136)

The authors indicated that the red fluorescence signal in the meristem region of the overexpressed transgenic sweetpotato root tip was significantly weaker than that of the wild-type. However, there is no such difference in red fluorescence signals between wild type and overexpressed transgenic lines. To me, overexpressed transgenic lines appear to have a stronger red fluorescence signal. Is it possible to indicate the intensity of the red fluorescence signal?

  1. Line 144

I think ‘Figure 7’ is wrong. It should be Figure 4.

  1. Line 146

Again, ‘Figure 7’ is wrong. It should be Figure 4.

  1. Lines 153-154

The authors indicated that there was no significant difference among the overexpressed transgenic lines, RNAi lines, or WT under normal growth conditions. However, from the pictures, I think that the leaves of the RNAi plants turned yellow even under normal conditions. Please make sure.

  1. Line: 177

I think ‘Figure 8’ is wrong, it should be Figure 5.

  1. Line: 186

In this section, the authors investigated the expression levels of several genes involved in different pathways, such as the ROS-scavenging system and the ABA signaling pathway. Here, the late embryogenesis abundant protein (LEA) gene suddenly appears. Why did the authors focus on this

gene? Please explain why you selected this gene and describe the relation of this gene to salt tolerance.

  1. Line 188

I think ‘Figure 9A’ is wrong, it should be Figure 6A.

  1. Line 197

I think ‘Figure 9B’ is wrong, it should be Figure 6B.

  1. Line 200

Again, ‘Figure 9’ should be wrong, it should be Figure 6.

  1. Line 235

Again, ‘Figure 9A, B’ should be wrong, it should be Figure 6A, B.

  1. Line 236

I think ‘Figure 10’ is wrong, should be Figure 7.

  1. Line 238

Again, I think ‘Figure 10’ is wrong, should be Figure 7.

  1. Line 239

The authors indicated that “↑Indicates up-regulation of gene coding these enzymes”. I think “red arrows indicate up-regulation of gene coding these enzymes” is better. In addition, since there are no down-regulated genes in this figure, please delete a sentence regarding down-regulated genes.

  1. Lines 287-288

I think this sentence is grammatically incorrect. Please make sure.

  1. Line 288

I could not access this URL. Please confirm.

  1. Line 290

Here, the authors should indicate ‘pI’ instead of ‘PI’. Please confirm.

  1. Lind 306

A space is required after the restriction enzyme name. please confirm.

  1. lines 314-315

I think this sentence is grammatically incorrect. Please make sure.

  1. lines 314-318

I don’t understand how to make overexpressed and RNAi transgenic plants from those sentences. For overexpressed plants, did you use a 35S promoter? How about RNAi plants?

Please describe more details about the method for making transgenic plants.

  1. Lines 328-330.

Please describe more details about the PI staining method. Or please provide a citation of this technique.

  1. Figure 1

I think the resolution of all figures (Figures 1A, B and C) is low. It is difficult for us to distinguish words such as amino acid names (Figure 1A) and species names (Figures 1B and 1C). Please indicate those words more clearly. In Figure 1D, do the blue boxes represent UTR regions (5’ UTR and 3’ UTR)? If so, please indicate.

  1. Figure 2A

What does the alphabet mean? In addition, please indicate the number of replicates (n=3?) for this experiment.

  1. Figure 3

I think this figure is strange. Are pictures on the top row correct? I don't think the authors can get the rightmost picture by merging the two pictures on the left. Also, please provide more explanation for this figure in the caption. Figure 3 is not easy to understand.

  1. Figure 9

The authors investigated CAT for overexpressed and DREB for RNAi. Why different?

  1. Figure 10

Is it possible to propose this model from the results of this research? Does ABA activate IbAPX, IbPOD, IbLEA and IbCAT genes? Are there any experimental results or previous research to support this? I think IbCAR1 may up-regulate IbAPX, IbPOD, IbLEA and IbCAT genes. Please consider this possibility.

  1. Supplementary Information

Figure S1. The word ‘Electophoresis’ should be wrong. Electrophoresis is correct.

Figure S2. (C) selection of Hyg-resistent embryonic callus. The word ‘Hyg-resistent’ should be wrong, Hyg-resistant is correct.

Figure S3. (B) selection of resistent callus. The word ‘resistent’ should be wrong, resistant is correct.

Author Response

Response to Reviewer 1

Dear Editors and Reviewers:

Thank you very much for your comments concerning our manuscript entitled “A C2-domain abscisic acid-related gene, IbCAR1, positively enhances salt tolerance in sweetpotato(Ipomoea batatas (L.) Lam.)” (ID: ijms-1847894). The comments are very valuable and helpful for revising and improving our paper, as well as the important guiding to our research. We have studied every comment carefully and have made correction or supplement one by one. The main correction in the manuscript and the responds to the Reviewer’s comments are as follows. Revised portion are marked with red color in the manuscript.

Point 1: In introduction (line: 56), the authors used the word ‘grain’ for sweetpotato. I think ‘crop’ is better for sweetpotato.

Response 1: It is really true as reviewer suggested, that we use the word ‘crop’ instead of ‘grain’ .

Point 2: In results (lines: 134-136), the authors indicated that the red fluorescence signal in the meristem region of the overexpressed transgenic sweetpotato root tip was significantly weaker than that of the wild-type. However, there is no such difference in red fluorescence signals between wild type and overexpressed transgenic lines. To me, overexpressed transgenic lines appear to have a stronger red fluorescence signal. Is it possible to indicate the intensity of the red fluorescence signal?

Response 2: The red fluorescence signal was mainly observed in meristematic zone of root tip. After treated with salt, the staining of the meristematic zone of root tip was observed with a fluorescence microscope Olympus BX 63. The results indicate that the red fluorescence signal in the meristem region of the overexpressed transgenic sweetpotato root tip was significantly weaker than that of the wild-type Xuzishu8.

Point 3: In line 144, I think ‘Figure 7’ is wrong. It should be Figure 4.

Response 3: It is really true as reviewer suggested, that we replace Figure 7 with Figure 4.

Point 4: In line 146, I think ‘Figure 7’ is wrong. It should be Figure 4.

Response 4: It is really true as reviewer suggested, that we replace Figure 7 with Figure 4.

Point 5: In lines 153-154, the authors indicated that there was no significant difference among the overexpressed transgenic lines, RNAi lines, or WT under normal growth conditions. However, from the pictrues, I think that the leaves of the RNAi plants turned yellow even under normal conditions. Please make sure.

Response 5: The 5 RNAi transgenic lines obtained in the later period were hydroponically treated, and it was found that under normal growth conditions, phenotypic results showed that there was no significant difference between RNAi transgenic lines and overexpressed transgenic lines and WT. Then we change the phenotypic pictrues.

Point 6: In line 177, I think ‘Figure 8’ is wrong, it should be Figure 5.

Response 6: It is really true as reviewer suggested, that we replace Figure 8 with Figure 5.

Point 7: In line 186, In this section, the authors investigated the expression levels of several genes involved in different pathways, such as the ROS-scavenging system and the ABA signaling pathway. Here, the late embryogenesis abundant protein (LEA) gene suddenly appears. Why did the authors focus on this gene? Please explain why you selected this gene and describe the relation of this gene to salt tolerance.

Response 7: We selected this gene because late embryogenesis abundant (LEA) proteins play an important role in plant growth and response to abiotic stresses. LEA proteins are critical in helping plants cope with salt stress. Yang identified a “Y1805”-specific LEA gene that was expressed highly and sensitively under salt stress using transcriptome analysis. It has been reported that late embryogenesis abundant gene LEA (Gh_A08G0694) enhances drought and salt stress tolerance in cotton.

Point 8: In line 188, I think ‘Figure 9A’ is wrong, it should be Figure 6A.

Response 8: It is really true as reviewer suggested, that we replace Figure 9A with Figure 6A.

Point 9: In line 197, I think ‘Figure 9B’ is wrong, it should be Figure 6B.

Response 9: It is really true as reviewer suggested, that we replace Figure 9B with Figure 6B.

Point 10: In line 200, I think ‘Figure 9’ is wrong, it should be Figure 6.

Response 10: It is really true as reviewer suggested, that we replace Figure 9 with Figure 6.

Point 11: In line 235, I think ‘Figure 9A,B’ is wrong, it should be Figure 6A,B.

Response 11: It is really true as reviewer suggested, that we replace Figure 9A,B with Figure 6A,B.

Point 12: In line 236, I think ‘Figure 10’ is wrong, it should be Figure 7.

Response 12: It is really true as reviewer suggested, that we replace Figure 10 with Figure 7.

Point 13: In line 238, I think ‘Figure 10’ is wrong, it should be Figure 7.

Response 13: It is really true as reviewer suggested, that we replace Figure 10 with Figure 7.

Point 14: In line 239, The authors indicated that “↑Indicates up-regulation of gene coding these enzymes”. I think “red arrows indicate up-regulation of gene coding these enzymes” is better. In addition, since there are no down-regulated genes in this figure, please delete a sentence regarding down-regulated genes.

Response 14: As Reviewer suggested that we replace the sentence “↑Indicates up-regulation of gene coding these enzymes” with “red arrows indicate up-regulation of gene coding these enzymes”, and we delete a sentence regarding down-regulated genes.

 Point 15: In lines 287-288, I think this sentence is grammatically incorrect. Please make sure.

 Response 15: We checked for syntax problems with this sentence. It has been modified into the following sentence. According to the EST obtained in previous study and referring to the genomic data of Ipomoea trifida (http://sweetpotato.uga.edu/), the cDNA sequence of IbCAR1 gene was obtained with a primer pair (IbCAR1-clon-F/R) and homology-based cloning method.

Point 16: In line 288, I could not access this URL. Please confirm.

Response 16: We have verified the URL.

Point 17: In line 290, here, the authors should indicate ‘pI’ instead of ‘PI’. Please confirm.

Response 17: It is really true as reviewer suggested, that we replace ‘PI’ with ‘pI’.

Point 18: In line 306, a space is required after the restriction enzyme name. please confirm.

Response 18: As reviewer suggested that we check the format.

Point 19: In lines 314-315, I think this sentence is grammatically incorrect. Please make sure.

Response 19: As reviewer suggested that we check the syntax.

Point 20: In lines 314-318, I don’t understand how to make overexpressed and RNAi transgenic plants from those sentences. For overexpressed plants, did you use a 35S promoter? How about RNAi plants? Please describe more details about the method for making transgenic plants.

Response 20: It is our negligence for it, which the method for making transgenic plants should be detailed. For overexpressed plants and RNAi plants, we use the 35S promoter to initiate gene transcription. We re-edited the method for making transgenic plants. It has been modified into the following sentence. The coding region of IbCAR1 was inserted into the pCAMBIA1301S expression vector. A pair of forward and reverse nonconserved fragments of IbCAR1 were inserted into the plant RNA interference (RNAi) vector pFGC5941. Vectors were individually introduced into Xuzishu8 via A. tumefaciens-mediated transformation as previously described. The resistant callus in good condition on the selecting medium were transferred to MS medium containing 1 mg/L ABA and 200 mg/L Cef, and the light intensity (2500 Lux) induced the formation of somatic embryos. Matrue somatic embryos were cultrued for 1-2 months to obtain transgenic plants. The putative transgenic sweetpotato plants were identified by PCR analysis with qRT-PCR primers.

Point 21: In lines 328-330, Please describe more details about the PI staining method. Or please provide a citation of this technique.

Response 21: It is our negligence for it, which the method for PI staining should be detailed. PI staining method has been added to the article. It has been modified into the following sentence. The transgenic and wild-type Xuzishu8 seedlings were hydroponic for 3 days and treated with 150 mmol·L-1 NaCl, while the control group was treated with water. After 24 h, three groups of sweetpotato root tips with good growth status were selected from each line, and the root tip length was about 3 cm. The root tips were then washed twice with phosphate buffer saline (PBS) and stained with PI mixtrue containing 10 µL PI Staining Solution and 190 µL 1×Assay Buffer for 30 min. Then, the root tips were rinsed twice with PBS, and the staining of the meristematic zone of root tip was observed with a fluorescence microscope Olympus BX 63.

Point 22: In Figure 1, I think the resolution of all figures (Figures 1A, B and C) is low. It is difficult for us to distinguish words such as amino acid names (Figure 1A) and species names (Figures 1B and 1C). Please indicate those words more clearly. In Figure 1D, do the blue boxes represent UTR regions (5’ UTR and 3’ UTR)? If so, please indicate.

Response 22: It is our negligence for not indicating those words very clearly. Now we have replaced the pictrue with a clearer one. The blue boxes represent UTR regions (5’ UTR and 3’ UTR). It has been noted in the article.

Point 23: In Figure 2A, what does the alphabet mean? In addition, please indicate the number of replicates (n=3?) for this experiment.

Response 23: Different lowercase letters indicate a significant difference at P < 0.05 based on Student’s t-test. Data are presented as means ± SE (n = 3). It has been noted in the article.

Point 24: In Figure 3, I think this figure is strange. Are pictrues on the top row correct? I don't think the authors can get the rightmost pictrue by merging the two pictrues on the left. Also, please provide more explanation for this figure in the caption. Figure 3 is not easy to understand.

Response 24: It is our negligence for choosing the wrong pictrue by mistake. Now, the top row of images has been modified.

Point 25: In Figure 9, The authors investigated CAT for overexpressed and DREB for RNAi. Why different?

Response 25: We wanted to double measure the expression levels of resistance-related genes in overexpressed and RNAi transgenic plants. We investigated DREB for overexpressed earlier, but the results were not ideal. In order to pursue a one-to-one correspondence between the results, now we present the pictrue of CAT instead of the pictrue of DREB for RNAi transgenic lines. Therefore, in this article, we will not show the expression of DREB in RNAi transgenic plants.

Point 26: In Figure 10, Is it possible to propose this model from the results of this research? Does ABA activate IbAPX, IbPOD, IbLEA and IbCAT genes? Are there any experimental results or previous research to support this? I think IbCAR1 may up-regulate IbAPX, IbPOD, IbLEA and IbCAT genes. Please consider this possibility.

Response 26: Yes, there are some previous research to support this point. ABA has been reported to regulate the expression levels of stress-tolerance-related genes, including APX, POD, LEA, and CAT in several plant species [49-51]. There are some previous research to support this. We have included these references in the article.

Point 27: In Supplementary Information, Figure S1. The word ‘Electophoresis’ should be wrong, Electrophoresis is correct. Figure S2. (C) selection of Hyg-resistent embryonic callus. The word ‘Hyg-resistent’ should be wrong, Hyg-resistant is correct. Figure S3. (B) selection of resistent callus. The word ‘resistent’ should be wrong, resistant is correct.

Response 27: We are very sorry for the misspelling of the data. We have corrected the spelling of total article.

Thank you again for your comments.

About submitting IJMS, I would like to make changes in the following areas.

The author's first name was wrong when uploading the system. We want to replace the Author's name “Han Wei Song” with “Weihan Song”. We want to replace the Author's name “Fei Run Gao” with “Runfei Gao”. We want to replace the Author's name “ Gang Yun Zhang” with “Yungang Zhang”. Revised portion are marked with red color in the manuscript.

With best regards,

Sincerely yours

Corresponding author: Qiang Li

Reviewer 2 Report

The manuscript entitled “A C2-domain abscisic acid-related gene, IbCAR1, positively enhances salt tolerance in sweetpotato(Ipomoea batatas (L.) 3 Lam.) is well written. However, few major corrections are needed before publishing in reputed journal. I recommend this manuscript to resubmit in the reputed journal after comprehensive corrections.

Authors are suggested to carefully check the spelling mistakes and abbreviation punctuation in the whole of the manuscript. For example;

Figure 8. C and G., replace “prolin” with “Proline)

Line no. 160., Replace “reactive oxygen species” with “ROS”.

The manuscript is based on the oxidative stress tolerance mechanism. However, the authors have not included the ROS data. Also, the authors have incorporated only SOD activity and added a few gene expressions to justify the involvement in the tolerance mechanism.

In figure 8H., MDA contents are lower in NaCl treatment as compared to WT. Is WT in a stress condition? Authors are suggested to justify.

In figure 9B., DREB shows >50% variation in expression level. Was the expression of DREB less?   

Author Response

Dear Editors and Reviewers:

Due to the network, the second revision reply was not submitted to the system. We will attach the second revision reply file as follows. The third modification reply file is submitted in the form of attachment, please check the attachment.

Response to Reviewer 2

Dear Editors and Reviewers:

Thank you very much for your comments concerning our manuscript entitled “A C2-domain abscisic acid-related gene, IbCAR1, positively enhances salt tolerance in sweetpotato(Ipomoea batatas (L.) Lam.)” (ID: ijms-1847894). The comments are very valuable and helpful for revising and improving our paper, as well as the important guiding to our research. We have studied every comment carefully and have made correction or supplement one by one. The main correction in the manuscript and the responds to the Reviewer’s comments are as follows. Revised portion are marked with red color in the manuscript.

Point 1: In figure 8C and G, replace “prolin” with “Proline”

Response 1: It is really true as reviewer suggested, that we replace “prolin” with “Proline”.

Point 2: In line 160, replace “reactive oxygen species” with “ROS”.

Response 2: It is really true as reviewer suggested, that we replace“reactive oxygen species”with“ROS”.

Point 3: The manuscript is based on the oxidative stress tolerance mechanism. However, the authors have not included the ROS data. Also, the authors have incorporated only SOD activity and added a few gene expressions to justify the involvement in the tolerance mechanism.

Response 3: The point of this article is not focused on the oxidative stress tolerance mechanism. Therefore, I chose four aspects of physiological indicators to measure, such as the oxygen reduction pathway, secondary metabolite pathway, membrane lipid peroxidation index, and ABA content. We measured ROS data to determine whether this gene was associated with the ROS scavenging pathway. So when we measured ROS data, we incorporated only SOD activity and added a few gene expressions to justify the involvement in the ROS signaling pathway.

Point 4: In figure 8H, MDA contents are lower in NaCl treatment as compared to WT. Is WT in a stress condition? Authors are suggested to justify.

Response 4: Yes, WT is in a stress condition. We repeated the experiment and re-measured the MDA content of wild-type Xuzizhu8 before and after salt stress treatment, and the results were consistent with the previous results.

Point 5: In figure 9B, DREB shows >50% variation in expression level. Was the expression of DREB less?   

Response 5: The expression of DREB was significantly reduced. We wanted to double measure the expression levels of resistance-related genes in overexpressed and RNAi transgenic plants. We investigated DREB for overexpressed earlier, but the results were not ideal. In order to pursue a one-to-one correspondence between the results, now we present the picture of CAT instead of the picture of DREB for RNAi transgenic lines. Therefore, in this paper, we will not show the expression of DREB in RNAi transgenic plants.

Thank you again for your comments.

About submitting IJMS, I would like to make changes in the following areas.

The author's first name was wrong when uploading the system. We want to replace the Author's name “Han Wei Song” with “Weihan Song”. We want to replace the Author's name “Fei Run Gao” with “Runfei Gao”. We want to replace the Author's name “ Gang Yun Zhang” with “Yungang Zhang”. Revised portion are marked with red color in the manuscript.

With best regards,

Sincerely yours

Corresponding author: Qiang Li

Reviewer 3 Report

The manuscript entitled "A C2-domain abscisic acid-related gene, IbCAR1, positively enhances salt tolerance in sweetpotato (Ipomoea batatas (L.) Lam.)" by You et al., was reviewed for publication in IJMS (manuscript 1847894). The manuscript examines the function of the gene IbCAR1, in the tolerance to salt stress of sweetpotato. While there is interesting data presented, the manuscript appears incomplete, the writing is poor, and the presentation of the results is marginal for some of the Figures. My comments below may help the authors improve the manuscript.

Lines 13 and 65, I have no idea what Xuzishu8 is, so not clear in the abstract and introduction sections.

Line 56, I don’t think sweet potato is an important “grain” crop.

Lines 72-80 Maybe it would be enough to indicate which sequence is the most similar to IbCAR1, and not list all the sequences given in Figure 1B in the text.

Line 94, I do not know what is meant by working site?

Figure 2B, I don’t follow or understand the explanation of this experiment at all. I am not clear on what exactly the controls are for these experiments.

Figure 3, These images do not appear very convincing, need to at least add arrows to indicate location of IbCAR1.

Lines 115-116, Where are the OsMADS53 and At-SYP122 controls?

Line 144, It appears that Figures 4-6 are missing, or the Figures are not correctly numbered.

Figure legend 7 needs to be improved to help interpret the images, and to complement the text in the results.

Author Response

Response to Reviewer 3

Dear Editors and Reviewers:

Thank you very much for your comments concerning our manuscript entitled “A C2-domain abscisic acid-related gene, IbCAR1, positively enhances salt tolerance in sweetpotato(Ipomoea batatas (L.) Lam.)” (ID: ijms-1847894). The comments are very valuable and helpful for revising and improving our paper, as well as the important guiding to our research. We have studied every comment carefully and have made correction or supplement one by one. The main correction in the manuscript and the responds to the Reviewer’s comments are as follows. Revised portion are marked with red color in the manuscript.

Point 1: In Lines 13 and 65, I have no idea what Xuzishu8 is, so not clear in the abstract and introduction sections.

Response 1: As Reviewer suggested that we introduce it in the Materials and Methods section.

Point 2: Line 56, I don’t think sweet potato is an important “grain” crop. 

Response 2: It is really true as reviewer suggested, that we use the word‘crop’  instead of‘grain’.

Point 3: In Lines 72-80, Maybe it would be enough to indicate which sequence is the most similar to IbCAR1, and not list all the sequences given in Figure 1B in the text.

Response 3: We entered IbCAR1 amino acid sequence into NCBI database for Blast analysis, and selected amino acid sequences of species with high homology in the alignment results. The results are shown in Figure 1B.

Point 4: In Line 94, I do not know what is meant by working site?

Response 4: As Reviewer suggested that we replace the sentence “To investigate the potential working site of IbCAR1 in Xuzishu8, we analyzed its expression levels in different tissues” with “To investigate the expression level of IbCAR1 in different tissues of Xuzishu8 ”.

Point 5: In Figure 2B, I don’t follow or understand the explanation of this experiment at all. I am not clear on what exactly the controls are for these experiments.

Response 5: It is our negligence for not indicating those sentence very clearly. Now we have replaced the sentence with a clearer one. At the same time, we give a more detailed explanation in “4.3. Expression Analysis of IbCAR1” of the Materials and Methods section.

Point 6: In Figure 3, These images do not appear very convincing, need to at least add arrows to indicate location of IbCAR1.

Response 6: As Reviewer suggested that we add arrows to indicate location of IbCAR1.

Point 7: Where are the OsMADS53 and At-SYP122 controls?

Response 7: The picture in the second column of Figure 3 are controled by the OsMADS53 and AtSYP122. We have indicated this in the picture note.

Point 8: In Line 144, It appears that Figures 4-6 are missing, or the Figures are not correctly numbered.

Response 8: It is really ture as reviewer suggested, that we change the Figures number.

Point 9: In Figure legend 7 needs to be improved to help interpret the images, and to complement the text in the results.

Response 9: As Reviewer suggested that we add a legend to Figure 7.

Thank you again for your comments.

About submitting IJMS, I would like to make changes in the following areas.

The author's first name was wrong when uploading the system. We want to replace the Author's name “Han Wei Song” with “Weihan Song”. We want to replace the Author's name “Fei Run Gao” with “Runfei Gao”. We want to replace the Author's name “ Gang Yun Zhang” with “Yungang Zhang”. Revised portion are marked with red color in the manuscript.

With best regards,

Sincerely yours

Corresponding author: Qiang Li

Round 2

Reviewer 2 Report

The authors have incorporated all the comments, accordingly. So, I recommend this manuscript to publish in its current form.

Reviewer 3 Report

A revised version of the manuscript entitled "A C2-domain abscisic acid-related gene, IbCAR1, positively enhances salt tolerance in sweetpotato (Ipomoea batatas (L.) Lam.)" by You et al., was reviewed for publication in IJMS (manuscript 1847894).

The authors made some corrections in response to the reviewer’s comments, but there are still problems with the manuscript, and some of the “corrections” added new errors, or were fairly minimal. There are numerous areas in the legends for the figures that are too minimal to interpret the data presented without referring to the methods section. A minimal amount of information in the legend needs to be included for the Figures to be stand-alone understandable. Throughout the manuscript there are gene names that are undefined. Finally, some of the data presented is not very convincing, and there are over interpretations of the data.

Line 13, Xuzishu8 is still not defined, and I think it should be clarified in the abstract. What is this a genotype or line or variety of sweet potato? Please clarify what this is when first used in the abstract and introduction, even if it is described more in the methods.

Lines 14-15, strongly induced? Based on the data presented in Figure 2B, IbCAR1 is induced by NaCL 3-fold after 12 hours, and around 3.7-fold by ABA after 48 hours, which to me is not “strongly induced”.

Line 17, not sure what “more complete” means, and the data that this is based on in Figure 4 is not very convincing.

Figure 2B legend, Why in Figure 2B graph in the legend, NaCl and ABA are indicated twice?

Lines 95-96, This corrected phrase is incomplete, so it is not corrected.

Line 111, significant is misspelled.

Figure 3, I still find this data not very convincing.

Lines 145-149, I am not convinced by the interpretation, “PI” is not defined in the text, the Figure is hard to interpret because there is not enough information in the legend to assist the reader.

Figure 6, The Figure is hard to interpret because there is not enough information in the legend to assist the reader. What are L1, L5 and L16, overexpression lines? What are #10 and, RNAi transgenic lines?

Lines 191-192, (APX, POD, and CAT), these need to be defined the first time used in the text.

Author Response

Response to Reviewer 3

Thank you very much for your comments concerning our manuscript entitled “A C2-domain abscisic acid-related gene, IbCAR1, positively enhances salt tolerance in sweetpotato(Ipomoea batatas (L.) Lam.)” (ID: ijms-1847894). The comments are very valuable and helpful for revising and improving our paper, as well as the important guiding to our research. We have studied every comment carefully and have made correction or supplement one by one. The main correction in the manuscript and the responds to the reviewer’s comments are as follows. Revised portion are marked with red color in the manuscript.

Point 1: In Line 13, Xuzishu8 is still not defined, and I think it should be clarified in the abstract. What is this a genotype or line or variety of sweet potato? Please clarify what this is when first used in the abstract and introduction, even if it is described more in the methods.

Response 1: As reviewer suggested that we add the introduction of Xuzishu8. It has been modified into the following sentence in the abstract and introduction.

“In this study, we cloned the IbCAR1 by homologous cloning method from the transcriptomic data of Xuzishu8, which is a sweetpotato cultivar with dark-purple flesh.”

Point 2: In Lines 14-15, strongly induced? Based on the data presented in Figure 2B, IbCAR1 is induced by NaCl 3-fold after 12 hours, and around 3.7-fold by ABA after 48 hours, which to me is not “strongly induced”.

Response 2: As reviewer suggested that we delete the “strongly”. It has been modified into the following sentence.

“This gene was expressed in all tissues of sweetpotato, with the highest expression level in leaf tissue and it could be induced by NaCl and ABA.”

Point 3: In Line 17, not sure what “more complete” means, and the data that this is based on in Figure 4 is not very convincing.

Response 3: As reviewer suggested that we modify this sentence. It has been modified into the following sentence.

“The PI staining experiment revealed the distinctive root cell membrane integrity of overexpressed transgenic lines upon salt stress.”

Point 4: In Figure 2B legend, Why in Figure 2B graph in the legend, NaCl and ABA are indicated twice?

Response 4: On the basis of the bar chart, we added the line chart to reflect the change trend after induction. So in Figure 2B graph in the legend, NaCl and ABA were labeled twice. As reviewer suggested that we modify the legend of Figure 2B. Now in the legend, NaCl and ABA were labeled only once.

Point 5: In Lines 95-96, This corrected phrase is incomplete, so it is not corrected.

Response 5: As Reviewer suggested that we replace the sentence “ 5’ UTR and 3’ UTR are represented by blue boxes.” with “ The blue boxes represent the 5’and 3’ untranslated region (UTR).”

Point 6: In Line 111, significant is misspelled.

Response 6: We have fixed the spelling of the word.

Point 7: In Figure 3, I still find this data not very convincing.

Response 7: Since it is difficult to achieve subcellular localization in sweetpotato, tobacco cells are commonly used for subcellular localization. The method was also used for subcellular localization of sweet potato in the following literatures.

  1. Zhang, H.; Gao, X.R.; Zhi, Y.H.; Li, X.; Zhang, Q.; Niu, J.B.; Wang, J.; Zhai, H.; Zhao, N.; Li, J.G.; Liu, Q.C.; He, S.Z. A non-tandem CCCH-type zinc-finger protein, IbC3H18, functions as a nuclear transcriptional activator and enhances abiotic stress tolerance in sweet potato. New Phytol. 2019, 223, 1918-1936.
  2. Kang, C.; Zhai, H.; Xue, L.Y.; Zhao, N.; He, S.Z.; Liu, Q.C. A lycopene beta-cyclase gene, IbLCYB2, enhances carotenoid contents and abiotic stress tolerance in transgenic sweetpotato. Plant Sci2018272, 243-254.

Point 8: In Lines 145-149, I am not convinced by the interpretation, “PI” is not defined in the text, the Figure is hard to interpret because there is not enough information in the legend to assist the reader.

Response 8: As reviewer suggested that we modify the explanation in lines 145-149. It has been modified into the following sentence.

 “After treated with salt, the root elongation zone (2-3 mm from the tip) of the WT showed a strong red fluorescence in the nucleus of most cells, indicating that the plasma membrane integrity in this root region was damaged. However, the PI-stained nucleus in the same position of the overexpressed transgenic lines were substantially smaller than that of the WT under saline conditions. These results showed that WT is more sensitive to salinity stress than the overexpressed transgenic lines”.

Meanwhile, we also made some changes to the legend. As reviewer suggested that we define “PI” in the legend.

Point 9: In Figure 6, The Figure is hard to interpret because there is not enough information in the legend to assist the reader. What are L1, L5 and L16, overexpression lines? What are #10 and, RNAi transgenic lines?

Response 9: Yes, L1, L5 and L16 represent the overexpression lines. #10 and #11 represent the RNAi transgenic lines. It has been noted in the legend.

Point 10: In lines 191-192, (APX, POD, and CAT), these need to be defined the first time used in the text.

Response 10: As reviewer suggested that we define them for the first time in the article. It has been modified into the following sentence.

“Our data indicated that after 12 h of the salt treatment, the expression levels of the ROS scavenging-related genes IbAPX, IbPOD, IbCAT (encoding a ascorbate peroxidase, a peroxidase, a catalase, respectively), the late embryogenesis abundant gene IbLEA (encoding a late embryogenesis abundant protein), the ABA biosynthesis-related genes IbAAO and IbABA2 (encoding a ascorbic acid oxidase, a zeaxanthin epoxidase, respectively) were significantly up-regulated in the overexpressed transgenic plants compared with WT under salt stress.”

Revised portion are marked with red colour in the manuscript.

Thank you again for your comments.

With best regards,

Yours sincerely.

Corresponding author: Qiang Li

Round 3

Reviewer 3 Report

The authors have addressed my concerns and I have no further comments.